# First Contact: Unsupervised Human-Machine Co-Adaptation via Mutual Information Maximization

**Siddharth Reddy, Sergey Levine, Anca D. Dragan**
University of California, Berkeley
{sgr,svlevine,anca}@berkeley.edu

## Abstract

How can we train an assistive human-machine interface (e.g., an electromyography-based limb prosthesis) to translate a user's raw command signals into the actions of a robot or computer when there is no prior mapping, we cannot ask the user for supervision in the form of action labels or reward feedback, and we do not have prior knowledge of the tasks the user is trying to accomplish? The key idea in this paper is that, regardless of the task, when an interface is more intuitive, the user's commands are less noisy. We formalize this idea as a completely unsupervised objective for optimizing interfaces: the mutual information between the user's command signals and the induced state transitions in the environment. To evaluate whether this mutual information score can distinguish between effective and ineffective interfaces, we conduct a large-scale observational study on 540K examples of users operating various keyboard and eye gaze interfaces for typing, controlling simulated robots, and playing video games. The results show that our mutual information scores are predictive of the ground-truth task completion metrics in a variety of domains, with an average Spearman's rank correlation of $\rho = 0.43$. In addition to offline evaluation of existing interfaces, we use our unsupervised objective to learn an interface from scratch: we randomly initialize the interface, have the user attempt to perform their desired tasks using the interface, measure the mutual information score, and update the interface to maximize mutual information through reinforcement learning. We evaluate our method through a small-scale user study with 12 participants who perform a 2D cursor control task using a perturbed mouse, and an experiment with one expert user playing the Lunar Lander game using hand gestures captured by a webcam. The results show that we can learn an interface from scratch, without any user supervision or prior knowledge of tasks, with less than 30 minutes of human-in-the-loop training.

## 1 Introduction

Imagine communicating with an intelligent extraterrestrial for the first time. They are trying to get us to do something (e.g., synthesize a novel molecule and build a space elevator out of it), but we do not know what that task is, and we cannot understand their language. They might speak, write, wave their limbs at us in a video, or send us messages through some other modality. They might even be watching how we act on their messages, and adapting the content of their messages to our behavior. How do we translate their messages into the actions they want us to take?[1]

To make progress on this problem in the absence of (known) alien signals, we study the related problem of helping humans communicate their intent to machines via arbitrary command signals; for example, controlling a robotic arm through a brain-computer interface [1]. When designing human-machine interfaces, we typically either (a) use human ingenuity to design an interface that is

---

Code, data, and videos available at `https://sites.google.com/view/coadaptation`

36th Conference on Neural Information Processing Systems (NeurIPS 2022).

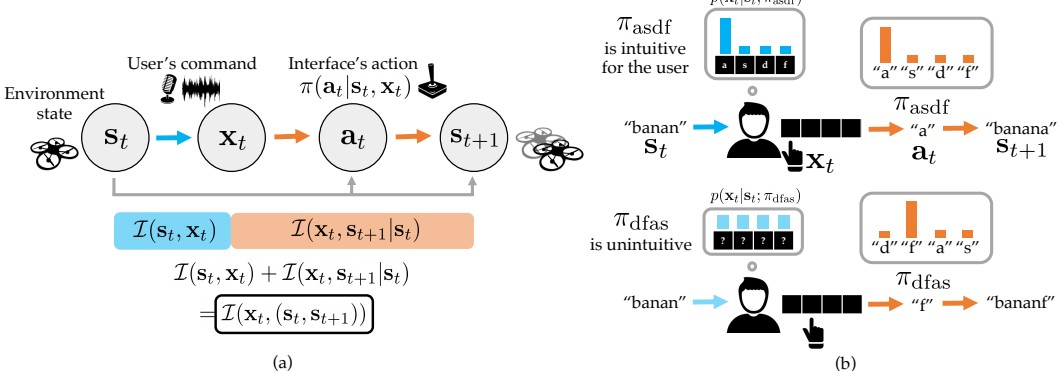

Figure 1: **(a)** The user is trying to perform a task (e.g., land a quadrotor), which the interface $\pi$ cannot directly observe. The user communicates with the interface $\pi$ by observing the environment state $\mathbf{s}_t$, then providing a command signal $\mathbf{x}_t$ (e.g., the raw audio of a voice command). The interface $\pi$ observes the state $\mathbf{s}_t$ and command $\mathbf{x}_t$, then takes an action $\mathbf{a}_t$ that causes the next state $\mathbf{s}_{t+1}$. We assume that the user partially adapts their command $\mathbf{x}_t$ to any given interface $\pi$, and that the extent of this user adaptation depends on the intuitiveness of the interface. We search for an interface $\pi$ that maximizes the rate of information transfer from user to the environment (orange) and from the environment to the user (blue), which we formalize as the mutual information $\mathcal{I}(\mathbf{x}_t, (\mathbf{s}_t, \mathbf{s}_{t+1}))$. **(b)** The more intuitive the interface (e.g., a QWERTY keyboard where the 'a', 's', 'd', and 'f' keys are arranged in that order), the less noisy the user's commands (e.g., keypresses). The less intuitive the interface (e.g., a randomly-permuted keyboard where "asdf" is scrambled to "dfas"), the noisier the user.

intuitive for a human to operate; (b) get direct supervision on how commands should map to actions, so we can train the interface through supervised learning; or (c) get reward feedback on the system's performance, so that we can improve the interface through reinforcement learning (see Sec. 4 for prior work). Unfortunately, none of this is possible with alien users: we do not know what is intuitive for them; we cannot ask them to perform specific tasks so that we can collect labeled data for supervised learning; we do not know what tasks they are trying to accomplish; and we cannot understand any feedback they might give on the quality of our actions. Existing approaches are also inconvenient for human users who would like to perform highly-personalized tasks using non-traditional interfaces, such as controlling a vehicle through a microphone [2], but do not want to be interrupted and asked for supervision or feedback.[1]

Without any source of supervision or prior knowledge about the user's desired tasks, it seems impossible to design a good interface. Recent work proposes optimizing for the user's ability to control their environment [3]. This is useful, but there tend to be many different interfaces that enable the user to influence the environment, and most of them are unintuitive to operate (e.g., compare the QWERTY keyboard to a keyboard with randomly-permuted keybindings). How do we capture what is intuitive, not only when the user is a human, but also when the user is an alien whose prior over 'natural' interfaces may not overlap with those of a human? The key idea (and assumption) in this paper: the more intuitive the interface, the less noisy the user's commands (Fig. 1b). Thus, a good interface should not only maximize the user's ability to influence the environment, but also minimize the entropy of the user's commands. We show that this corresponds to maximizing the mutual information between the user's command and the induced state transition (Sec. 2.1).

Of course, we cannot optimize this mutual information objective without interacting with the user. What we thus propose is a co-adaptation process, wherein we start with a random interface, the user learns about it as they use it, we measure the mutual information, and update the interface to maximize the mutual information through reinforcement learning. As the interface improves, the user's ability to complete their desired tasks using the interface also increases. We call this method the *mutual information-maximizing interface* (MIMI).

---

[1]"'The van is under voice command,' [Ng] explains. 'I removed the steering-wheel-and-pedal interface because I found verbal commands more convenient. This is why I will sometimes make unfamiliar sounds with my voice – I am controlling the vehicle's systems...Ng makes a yapping sound, and the van pulls out onto the frontage road...Ng laughs sharply, like distant ack-ack, and the van almost swerves off the road...Ng sings a little song. A robot arm unfolds itself from the ceiling of the van, crisply yanks the vial from her hand, swings it around, and holds it in front of a video camera set into the dashboard." [2]

Our primary contribution is an unsupervised mutual information objective for evaluating interfaces that can be used even when the ground-truth reward function for the user's desired task is unknown. To evaluate whether this unsupervised objective can be used to distinguish between effective and ineffective interfaces, we analyze offline datasets of users operating various interfaces in five different domains from prior work, and measure the correlation between the ground-truth rewards for each task and our unsupervised mutual information scores (Sec. 5.1). The results show that, in each domain, the interface with the largest mutual information score also has the largest ground-truth reward (Fig. 5 in the appendix). We also contribute the MIMI algorithm for training an interface to understand human commands from scratch, without prior knowledge of the user's desired tasks or explicit feedback from the user. We evaluate MIMI through an online user study with 12 participants who use a perturbed mouse to perform a 2D cursor control task (Sec. 5.2). The results show that, in under 30 minutes of human-in-the-loop training, MIMI learns an interface that is intuitive for users to operate and enables users to reach their goals more quickly than a random interface. We also showcase MIMI's ability to scale to more complex tasks and command modalities through a study with one expert user (the first author) who uses hand gestures to play the Lunar Lander game (Sec. 5.3).

## 2 An Unsupervised Mutual Information Objective for Evaluating Interfaces

In our problem setting, users cannot directly act in the environment, and must rely on an interface to take actions for them (e.g., due to a motor impairment, or because they are remotely operating space robots, or because they are aliens communicating with humans over long distances). The interface does not have direct access to the user's desired task, but can observe command signals that the user provides to the interface (e.g., through a webcam or microphone). We formulate the command-to-action translation problem as a contextual Markov decision process (CMDP) [4]. Each observation is decomposed into two variables: the environment state $\mathbf{s}_t$ (e.g., the position and orientation of a robot arm) and the user's command signal $\mathbf{x}_t$ (e.g., a webcam image of their eye gaze). The interface $\pi(\mathbf{a}_t|\mathbf{s}_t, \mathbf{x}_t)$ takes an action $\mathbf{a}_t$ given the state $\mathbf{s}_t$ and command $\mathbf{x}_t$. We do not assume access to the user's desired task (which represents the 'context' in the CMDP), the space of desired tasks, the ground-truth reward function $R(\mathbf{s}_t, \mathbf{a}_t)$, or the state transition dynamics. We approach this problem by defining a surrogate reward function that correlates positively with the ground-truth reward. We define this surrogate to be the information rate of the interface, which we formalize as the mutual information between the user's command $\mathbf{x}_t$ and the induced state transition $(\mathbf{s}_t, \mathbf{s}_{t+1})$. Fig. 1a outlines our method.

### 2.1 Characterizing the Information Rate of the Interface

Viewing our system through the lens of information theory [5, 6], the user interacts with the environment through a noisy communication channel that mediates perception (blue in Fig. 1) and control (orange in Fig. 1). When the environment sends a message $\mathbf{s}_t$ to the user, the user's internal decision-making process noisily converts the message $\mathbf{s}_t$ into a command $\mathbf{x}_t$. When the user sends a message $\mathbf{x}_t$ to the interface, the interface converts the message $\mathbf{x}_t$ into an action $\mathbf{a}_t \sim \pi(\mathbf{a}_t|\mathbf{s}_t, \mathbf{x}_t)$, and the environment converts the action $\mathbf{a}_t$ into a next state $\mathbf{s}_{t+1} \sim p(\mathbf{s}_{t+1}|\mathbf{s}_t, \mathbf{a}_t)$ following unknown dynamics. In this section, we derive the information rate of this channel, and show how maximizing it enables the user to perform their desired tasks.

The amount of information that is transmitted from the user to the environment can be characterized as the conditional mutual information $\mathcal{I}(\mathbf{x}_t, \mathbf{s}_{t+1}|\mathbf{s}_t)$ (orange in Fig. 1). Maximizing this term enables the user to influence the next state $\mathbf{s}_{t+1}$ by modulating their command $\mathbf{x}_t$. However, there can be many different interfaces that achieve an equally-high value of $\mathcal{I}(\mathbf{x}_t, \mathbf{s}_{t+1}|\mathbf{s}_t)$. For example, consider typing on a QWERTY keyboard. Randomly permuting the keybindings would preserve the value of $\mathcal{I}(\mathbf{x}_t, \mathbf{s}_{t+1}|\mathbf{s}_t)$, since there would still be a one-to-one mapping between physical keys and typed characters, but the resulting layout would probably not be as intuitive to use as QWERTY. In order to maximize the intuitiveness of the interface, we need to consider the amount of information that gets transmitted in the other direction: from the environment to the user.

The amount of information that gets transmitted from the environment to the user can be characterized as the mutual information $\mathcal{I}(\mathbf{s}_t, \mathbf{x}_t)$ (blue in Fig. 1). The main assumption in our method is that maximizing $\mathcal{I}(\mathbf{s}_t, \mathbf{x}_t)$, which can be rewritten as $\mathcal{H}(\mathbf{x}_t) - \mathcal{H}(\mathbf{x}_t|\mathbf{s}_t)$, leads to an intuitive interface. The idea behind penalizing the conditional entropy $\mathcal{H}(\mathbf{x}_t|\mathbf{s}_t)$ is that, when the user operates an

**Algorithm 1** MIMI-EVALUATE($\pi$)

1: $\mathcal{D} \leftarrow \emptyset$
2: **while** $|\mathcal{D}| < k$ **do**
3:      $\mathbf{x}_t \sim p(\mathbf{x}_t|\mathbf{s}_t; \pi)$                  ▷ *user gives command that is partially adapted to interface $\pi$*
4:      $\mathbf{a}_t \sim \pi(\mathbf{a}_t|\mathbf{s}_t, \mathbf{x}_t)$                               ▷ *interface takes action*
5:      $\mathbf{s}_{t+1} \sim p(\mathbf{s}_{t+1}|\mathbf{s}_t, \mathbf{a}_t)$ ▷ *environment transitions to next state (following unknown dynamics)*
6:      $\mathcal{D} \leftarrow \mathcal{D} \cup \{(\mathbf{s}_t, \mathbf{x}_t, \mathbf{s}_{t+1})\}$
7: Split $\mathcal{D}$ into training set $\mathcal{D}_{\text{train}}$ and validation set $\mathcal{D}_{\text{val}}$
8: $\phi^*, \psi^* \leftarrow \arg\max_{\phi,\psi} I_{\text{TUBA}}(\phi, \psi; \mathcal{D}_{\text{train}})$            ▷ *optimize MI lower bound in Eqn. 2*
9: Return $I_{\text{TUBA}}(\phi^*, \psi^*; \mathcal{D}_{\text{val}})$      ▷ *return optimized MI lower bound evaluated on validation set*

unintuitive interface, they provide noisier commands $\mathbf{x}_t$ given the current state $\mathbf{s}_t$ (similar to the maximum causal entropy model of noisily-rational user behavior [7]). This is because an unintuitive interface can confuse users, cause them to provide random commands when they are under time pressure to perform tasks, or cause them to keep trying different commands to see which ones will trigger their desired actions. On the other hand, an intuitive interface makes it easy for the user to determine which command will induce their desired next state, leading to more deterministic user behavior (Fig. 1b). Penalizing the conditional entropy $\mathcal{H}(\mathbf{x}_t|\mathbf{s}_t)$ alone is not enough, however. For example, consider an interface that minimizes the conditional entropy to zero by frustrating the user and causing them to always provide the same command, no matter the state or desired task (e.g., a constant 'no op' signal). Hence, we must also maximize the marginal entropy $\mathcal{H}(\mathbf{x}_t)$, which leads to an interface that encourages the user to provide a wide distribution of commands overall (e.g., by enabling the user to visit a variety of states).

Summing the two mutual information terms, we get the total amount of information that the user sends and receives, which simplifies to the mutual information between the user's command $\mathbf{x}_t$ and the induced state transition $(\mathbf{s}_t, \mathbf{s}_{t+1})$,

$$
\begin{aligned}
R(\pi) &\triangleq \mathcal{I}(\mathbf{s}_t, \mathbf{x}_t) + \mathcal{I}(\mathbf{x}_t, \mathbf{s}_{t+1}|\mathbf{s}_t) \\
&= (\mathcal{H}(\mathbf{x}_t) - \mathcal{H}(\mathbf{x}_t|\mathbf{s}_t)) + (\mathcal{H}(\mathbf{x}_t|\mathbf{s}_t) - \mathcal{H}(\mathbf{x}_t|\mathbf{s}_t, \mathbf{s}_{t+1})) \\
&= \mathcal{H}(\mathbf{x}_t) - \mathcal{H}(\mathbf{x}_t|\mathbf{s}_t, \mathbf{s}_{t+1}) \\
&= \mathcal{I}(\mathbf{x}_t, (\mathbf{s}_t, \mathbf{s}_{t+1})).
\end{aligned}
\tag{1}
$$

## 2.2 Estimating Mutual Information Rewards

Computing the mutual information reward $R(\pi)$ exactly would require taking an expectation with respect to the state transition dynamics of the environment as well as the user's policy for giving commands, both of which are unknown. Hence, we estimate the mutual information by collecting samples $(\mathbf{s}_t, \mathbf{x}_t, \mathbf{s}_{t+1})$ of the user operating the interface $\pi$. Alg. 1 outlines this sample-based evaluation of $R(\pi)$.

To estimate $\mathcal{I}(\mathbf{x}_t, (\mathbf{s}_t, \mathbf{s}_{t+1}))$ from the samples in a dataset $\mathcal{D}$ in our experiments, we use the TUBA estimator [8, 9]. In particular, we fit a "statistics network" $T_\phi$ and variational parameters $a_\psi$ to maximize the mutual information lower bound,

$$
I_{\text{TUBA}}(\phi, \psi) \triangleq \sum_{(\mathbf{s}_t, \mathbf{x}_t, \mathbf{s}_{t+1}) \in \mathcal{D}} T_\phi(\mathbf{x}_t, \mathbf{s}_t, \mathbf{s}_{t+1}) - \mathbb{E}_{\bar{\mathbf{x}} \sim \mathcal{D}} \left[ \frac{e^{T_\phi(\bar{\mathbf{x}}, \mathbf{s}_t, \mathbf{s}_{t+1})}}{e^{a_\psi(\mathbf{s}_t, \mathbf{s}_{t+1})}} + a_\psi(\mathbf{s}_t, \mathbf{s}_{t+1}) - 1 \right], \quad (2)
$$

where $\bar{\mathbf{x}}$ is sampled uniformly at random from the dataset $\mathcal{D}$. Intuitively, $T_\phi$ is a discriminator that learns to distinguish between realistic pairs $(\mathbf{x}_t, (\mathbf{s}_t, \mathbf{s}_{t+1}))$ and unrealistic pairs $(\bar{\mathbf{x}}, (\mathbf{s}_t, \mathbf{s}_{t+1}))$. In our experiments, we represent $T_\phi$ and $a_\psi$ each as a feedforward neural network that outputs a scalar.

Two practical issues emerge when applying the standard TUBA estimator to our problem setting. First, we can only collect a small amount of data $\mathcal{D}$ to evaluate a given interface during our user studies, which can lead to high-variance estimates of the mutual information. To address this issue, we adopt the following approach from prior work on data-efficient mutual information estimation [10]: instead of using the final training loss $I_{\text{TUBA}}$ as our mutual information estimate, we use the validation loss (line 9 in Alg. 1). Second, we must avoid substantial wall-clock delays to the user

while we optimize the mutual information lower bound in line 8 of Alg. 1. Typically, in order to accurately estimate the mutual information, one must optimize the lower bound to convergence. This can take many steps of stochastic gradient descent, and blocks the user from proceeding to the next episode. In our setting, however, we find that optimizing to convergence is unnecessary. The key idea is that, since we ultimately use the mutual information estimates as rewards to compare interfaces, only the relative values of the estimates matter to us. Hence, we only take 1K gradient steps (with a small batch size of 64) to fit the estimator in all our experiments. We find that, even after only 1K steps, the mutual information estimates already tend to correlate positively with the ground-truth task rewards (Fig. 2).

## 3 Learning an Interface through Mutual Information Maximization

In principle, one could represent the interface $\pi$ in any manner (e.g., as a deep neural network) and use any reinforcement learning algorithm to optimize MIMI-EVALUATE($\pi$). In our experiments, we parameterize the interface as a linear model $\pi_\theta$, where $\theta$ are the weights and biases, and maximize MIMI-EVALUATE($\pi$) using an off-the-shelf Bayesian optimization algorithm (`scikit-optimize`) [11] based on Gaussian process regression [12]. Initially, we randomly sample the interface parameters $\theta_0$, have the user attempt to perform their desired tasks using the interface $\pi_{\theta_0}$ for 10 episodes, and compute the mutual information reward MIMI-EVALUATE($\pi_{\theta_0}$). Given all pairs of interface parameters and corresponding mutual information rewards $\{(\theta_i, \text{MIMI-EVALUATE}(\pi_{\theta_i}))\}_{i=0}^h$ evaluated so far, we fit a Gaussian process regression model that predicts the mutual information reward given the interface parameters, and select the next interface $\theta_{h+1}$ by optimizing one of three acquisition functions—expected improvement, lower confidence bound, or probability of improvement—chosen uniformly at random. Details in Appendix A.2.

## 4 Related Work

Developing an assistive human-machine interface from scratch requires translating the unknown language of commands into actions. Translating unknown languages is a broad area of research, encompassing linguistics [13], cryptanalysis [14], and deep neural network interpretability [15, 16, 17]. Prior work on multi-agent reinforcement learning proposes inductive biases that use mutual information maximization to promote positive signalling and listening in cooperative tasks [18, 19]. This prior work studies the emergence of communication in autonomous agents, whereas MIMI is intended for the unsupervised learning of co-adaptive user interfaces. Our problem setting also differs in that we cannot train the speaker agent (i.e., the human), so we must craft an objective for the listener agent (i.e., the interface) that indirectly promotes informative signaling from the speaker.

The literature on adaptive interfaces spans multiple fields, including brain-computer interfaces [20, 21, 22, 23, 24, 25, 26], natural language interfaces [27, 28], speech interfaces [29, 30], electronic musical instruments [31], and robotic teleoperation interfaces [32, 33, 34, 35, 3, 36]. In particular, there is substantial prior work on human-machine co-adaptation [37, 38, 39, 40, 41, 42, 43]. In contrast to this prior work, MIMI does not require knowledge of the user's desired tasks or direct supervision. [44, 45] propose an unsupervised co-adaptation method that uses principal component analysis to fit an interface that maps commands to actions, but it requires that the interface be a linear mapping, and it does not necessarily maximize the intuitiveness of the interface.

Prior work has trained reinforcement learning agents to optimize implicit user feedback contained in EEG signals [46], peripheral pulse measurements [47], facial expressions [48, 49], and clicks [50]. These approaches typically assume access to a dataset of user observations (e.g., facial expressions) and explicit rewards (e.g., labels that indicate positive or negative emotion) in order to train a reward model that infers rewards from user observations. Other prior work on human-in-the-loop RL assumes direct access to explicit user feedback during the RL phase [51, 32, 52, 53, 34]. MIMI, in contrast, does not require access to explicit rewards at any time.

Mutual information maximization has previously been used to acquire diverse, complex behaviors through reinforcement learning in the absence of extrinsic rewards [54, 55, 56, 57, 58, 59, 60]. These prior methods aim to train an autonomous agent, whereas MIMI aims to train a user interface. Furthermore, MIMI differs in that it learns from a user who adaptively provides commands to the interface that try to induce the user's desired next states, whereas in these prior methods, the 'command

signal' is a latent code that is randomly sampled from a fixed distribution. As a consequence, the latent skill space discovered by these prior methods is not necessarily intuitive for a user to control, or even human-interpretable.

Our mutual information objective in Eqn. 1 is related to empowerment [61], which measures the channel capacity between an agent's actions and the environment state. Empowerment has previously been used as an intrinsic motivation for autonomous reinforcement learning agents [62], and an auxiliary objective for assistive agents that preserve a user's ability to reach various states [3]. Our objective differs in that it measures not only the user's ability to influence state transitions, but also the intuitiveness of the interface.

Our implicit model of the user's noisy rationality in Sec. 2.1 is similar to the widely-used maximum causal entropy model of user behavior [7], but departs from it in that we assume the user partially adapts their behavior to match the interface, and that the extent of this user adaptation depends on the intuitiveness of the interface.

## 5 Experimental Evaluation

Our experiments focus on measuring MIMI's effectiveness at evaluating interfaces, learning interfaces from scratch, and scaling interface optimization to complex tasks and interface modalities where there is no clear, intuitive solution. We aim to answer the following questions. **Q1** (Sec. 5.1): Can MIMI's mutual information score distinguish between effective and ineffective interfaces for a variety of users, interface modalities, environments, and tasks? **Q2** (Sec. 5.2): Can we learn an interface from scratch in a completely unsupervised manner by maximizing mutual information rewards with MIMI? **Q3** (Sec. 5.3): Does unsupervised human-in-the-loop reinforcement learning scale to more complex interface modalities and tasks? To answer **Q1**, we conduct an observational study on five datasets from prior work. To answer **Q2**, we conduct a user study with 12 participants who use a perturbed mouse to perform a 2D cursor control task. To answer **Q3**, we study how MIMI can be used to enable an expert user (the first author) to play the Lunar Lander game through hand gestures captured by a webcam.

### 5.1 Offline Evaluation of Existing Interfaces

In this experiment, we aim to evaluate whether the mutual information score in MIMI can distinguish between effective and ineffective interfaces for a wide variety of users, interface modalities, environments, and tasks (**Q1**). We take data from prior work on adaptive interfaces in which the ground-truth rewards were measured, and check whether MIMI's unsupervised evaluation of those interfaces correlates with the true reward that users received when performing tasks via those interfaces. We examine data from four prior works: X2T [63], ASHA [64], shared autonomy via deep reinforcement learning (SAvDRL) [65], and internal-to-real dynamics transfer (ISQL) [66]. In each of these prior works, users were asked to operate at least two different interfaces: a non-adaptive baseline interface, and an adaptive interface that learns to assist users over time. In the X2T experiments, participants used their eye gaze (a 128-dimensional command signal $\mathbf{x}_t$ from their webcam) to select 1 of 8 buttons on a screen in order to type a phrase. In the ASHA experiments, participants used their eye gaze to control a 7-DoF simulated robotic arm to either flip a light switch or reach for a bottle. In the SAvDRL and ISQL experiments, participants used the directional keys on a keyboard to play the Lunar Lander game [67]. The appendix describes each domain in detail. The data from these prior user studies totals over 540K timesteps.

For each domain, each user, and each experimental condition (e.g., non-adaptive vs. adaptive interface), we measure our mutual information score by averaging over 10 different random seeds used to fit the estimator in Eqn. 2. The ground-truth reward function for each domain is defined in Appendix A.3. The results in Fig. 2 show that, in 4 out of 5 domains, MIMI's mutual information score for an interface is predictive of the ground-truth task reward a user is able to obtain using that interface, with an average Spearman's rank correlation coefficient of $\rho = 0.43$. Furthermore, in each domain, the interface with the largest mutual information score also has the largest ground-truth reward (Fig. 5 in the appendix).

In analyzing the data from the SAvDRL experiments, we initially encountered an unexpected result: a strong negative correlation between our mutual information scores and the ground-truth rewards

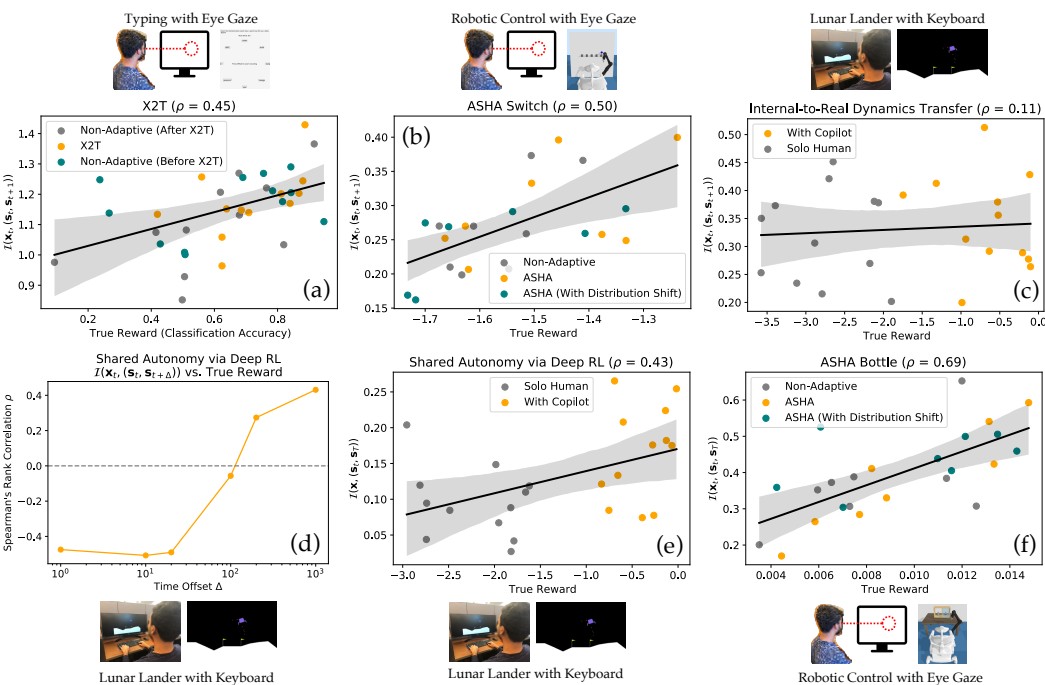

Figure 2: In 4 out of 5 domains, our mutual information score (y-axis) is predictive of the ground-truth reward (x-axis) that a user is able to obtain with a given interface. **(a)** In X2T, ground-truth rewards correspond to the classification accuracy of the interface, which predicts the user's desired button (1 of 8) given the user's eye gaze signals. There were three conditions in the X2T experiments: the user first operated the non-adaptive baseline interface, then operated the adaptive X2T interface, then returned to operating the non-adaptive baseline interface. There were 12 participants in the user study, yielding a total of $12 \cdot 3 = 36$ data points in the scatter plot (one for each user in each condition). Spearman's rank correlation coefficient $\rho = 0.45$ between the ground-truth rewards and the estimated mutual information scores. **(b)** Ground-truth rewards penalize distance from the robot end effector to the position of the target switch. **(c)** In the internal-to-real dynamics transfer data, there were only two conditions: the user playing the Lunar Lander game on their own, and with assistance. **(d)** In the SAvDRL data, the rank correlation between the ground-truth reward and the mutual information score depends on the time offset $\Delta$ in the generalized mutual information objective $\mathcal{I}(\mathbf{x}_t, (\mathbf{s}_t, \mathbf{s}_{t+\Delta}))$. **(e)** Following the results in (d), we set the time offset $\Delta$ to the maximum episode length of $10^3$. Ground-truth rewards penalize crashing, and give a bonus for landing between the flags. **(f)** Ground-truth rewards give a bonus for opening the door in front of the desired bottle and for reaching the desired bottle. As in (e), we set the time offset $\Delta$ to the maximum episode length.

(Fig. 2d). The reason is that, in the SAvDRL experiments, the assistant tends to help the user play the Lunar Lander game by preventing them from crashing, which necessarily involves ignoring a large fraction of the user's commands. This leads to a low mutual information between the user's command $\mathbf{x}_t$ and the one-step state transition $(\mathbf{s}_t, \mathbf{s}_{t+1})$. However, even though the assistant appears to reduce the user's influence over the next state, it actually increases the user's influence over later states in the episode by preventing the user from crashing immediately. This leads us to propose a generalized mutual information objective, $\mathcal{I}(\mathbf{x}_t, (\mathbf{s}_t, \mathbf{s}_{t+\Delta}))$, in which the time offset $\Delta$ is a hyperparameter than can be set to a value other than $\Delta = 1$.

Fig. 2d shows how modifying the time offset $\Delta$ in the generalized mutual information objective $\mathcal{I}(\mathbf{x}_t, (\mathbf{s}_t, \mathbf{s}_{t+\Delta}))$ affects the correlation between the mutual information scores and the ground-truth rewards in the SAvDRL experiments: for low values of $\Delta$ (including the default value of $\Delta = 1$), there is a negative correlation, whereas for sufficiently high values of $\Delta$ (i.e., the maximum episode length of $10^3$ steps), there is a positive correlation. In Figures 2e and 2f, we set $\Delta$ to the maximum episode length, so that the MIMI objective $\mathcal{I}(\mathbf{x}_t, (\mathbf{s}_t, \mathbf{s}_T))$ always measures mutual information with respect to the final state $s_T$ of each episode. In general, choosing the value of $\Delta$ requires prior knowledge of the timescale of the user's desired influence over the system (i.e., does the user's command indicate what should happen at the next timestep, or what should happen by the end of the episode?). This is the primary limitation on the generality of our method.

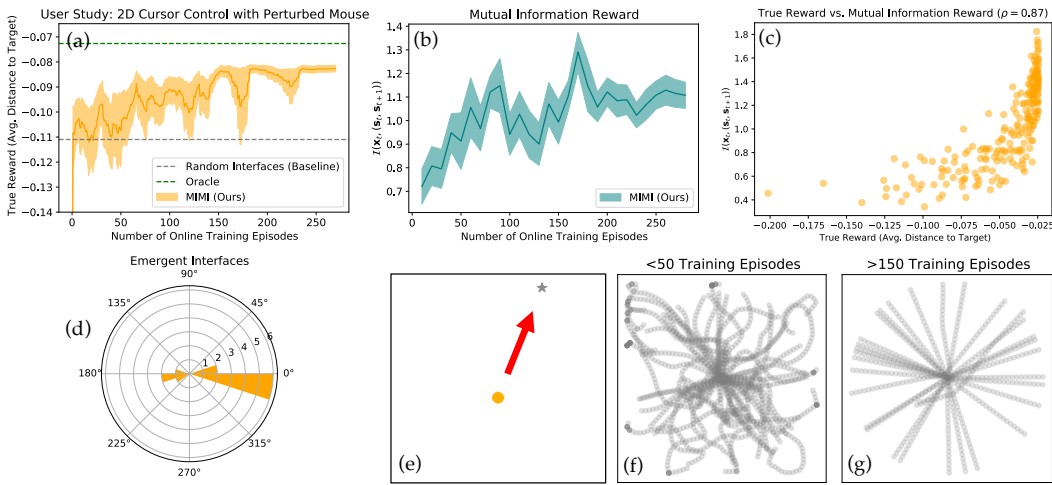

Figure 3: **(a)** Through online training, MIMI (orange) learns an interface that substantially outperforms baseline interfaces with random parameters (gray), and approaches the performance of an oracle agent that always moves straight to the target (green). **(b)** As expected, MIMI is indeed maximizing mutual information rewards through human-in-the-loop RL. **(c)** Each point in the scatter plot represents a distinct user operating a distinct interface. The ground-truth task rewards (negative average distance to target) and the mutual information rewards are highly positively correlated (Spearman's rank correlation $\rho = 0.87$), as in the offline experiments in Sec. 5.1. **(d)** A polar histogram of the final interface parameter $\theta$ that MIMI converges to, for each of the 12 users. Whereas the random baseline samples $\theta$ from a uniform distribution over perturbation angles $[0, 2\pi)$, MIMI converges to a highly non-uniform distribution over $\theta$. In particular, MIMI tends to converge to one of two interfaces: no perturbation of the user's mouse ($\theta \approx 0$), or inversion of the user's mouse ($\theta \approx \pi$). **(e-g)** In this center-out cursor control task, the user initially tends to move in curved trajectories that wander away from the target. After 150 episodes of online training, the user tends to move in straight lines to the target.

## 5.2 User Study: 2D Cursor Control with Perturbed Mouse

In the previous experiment, we showed that MIMI's mutual information score correlates positively with the ground-truth task rewards in a variety of offline datasets (**Q1**). However, it is not clear from this result alone that maximizing mutual information rewards will yield an interface that also performs well in terms of the ground-truth task rewards (**Q2**). To answer **Q2**, we conduct a small-scale user study with 12 participants who use a perturbed mouse to perform a simple 2D cursor control task. The goal of the task is to drive a cursor from the center of the screen to a randomly-sampled target position (Fig. 3e). The ground-truth reward is the average negative distance to target throughout the episode. The user's 2D mouse position command is used to control the 2D velocity of the cursor (details in Appendix A.3). The interface $\pi_\theta$ applies a rotation to the 2D mouse position to get the 2D velocity (i.e., $\theta$ consists of a single parameter: the rotation angle). The initial interface $\pi_{\theta_0}$ is a random rotation, and is usually difficult for the user to operate (see scattered trajectories in Fig. 3f). Users struggle to anticipate and adjust for the rotation, which leads to them initially moving away from the target, oscillating around the target as they approach it because they keep steering in the wrong direction, or getting stuck on the boundaries of the environment because they forget how to steer away from the boundary.

Each of the 12 participants completed two phases of experiments: A and B. In phase A, they operate 5 interfaces with randomly-sampled parameters $\theta$ for 10 episodes each. In phase B, they operate an adaptive interface optimized by MIMI (Sec. 3). Each interface proposed by MIMI is evaluated for 10 episodes, and we halt the experiment after at most 30 interfaces have been evaluated, yielding a total of at most 300 episodes per participant. To avoid the confounding effect of overall user improvement or fatigue over time, we counterbalance the order of phases A and B. Details in Appendix A.1.

The results in Fig. 3 show that, even though MIMI starts from scratch with a random interface and only aims to maximize mutual information rewards (blue), it enables users to achieve substantially higher ground-truth rewards (orange) than the random baseline interfaces (gray). Fig. 3d shows that MIMI tends to converge to one of two interfaces: moving the cursor in the same direction as the

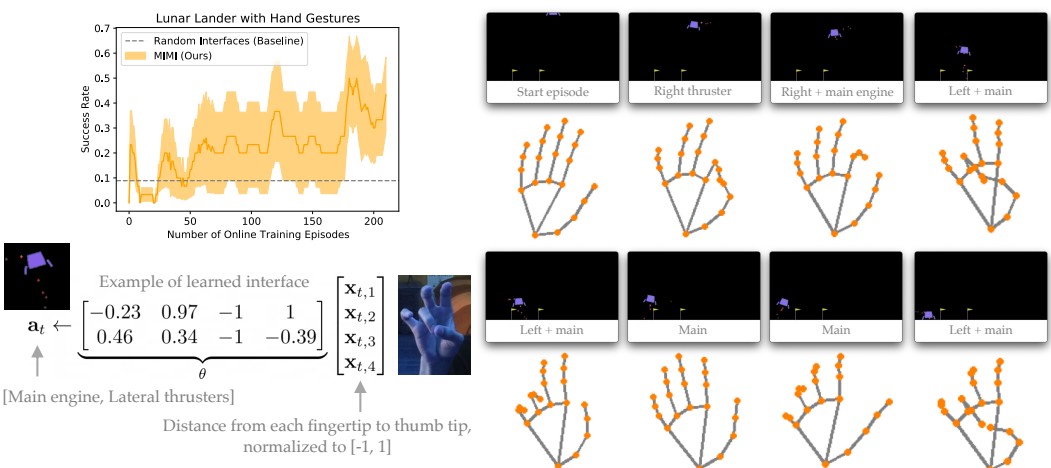

Figure 4: MIMI can learn an effective mapping from hand pose commands to thruster actions in the Lunar Lander game. During the initial training episodes when the interface is randomly sampled, the interface can require uncomfortable hand contortions in order to trigger certain actions, or have a very sharp threshold between poses that trigger one action vs. a different action. Towards the end of training, MIMI learns a comfortable, intuitive interface (one episode illustrated above) that enables occasional successful landings in a challenging game where constantly crashing is the default behavior. Success rates are averaged over 3 different random seeds, and smoothed using a moving average with a window size of 10 episodes. Hand tracking is performed using a webcam and MediaPipe [68]. See Fig. 6 in the appendix for plots of the mutual information reward. Videos available at https://sites.google.com/view/coadaptation.

mouse, or rotating the mouse direction 180 degrees (which tends to be easy for users to understand and invert; see the appendix).

## 5.3 Lunar Lander with Hand Gestures

In the previous experiment, we showed that MIMI is capable of learning an interface from scratch through human-in-the-loop RL with the mutual information objective, albeit on a simple task and interface modality (2D cursor control via 2D mouse commands). In this section, we demonstrate that this result extends to a more complex task and interface modality (**Q3**): an expert user (the first author) playing the Lunar Lander game [67] using hand gestures. To win the game, the user must land between the flags without crashing. The user's hand pose command is a 4D signal that consists of the distances from the thumb tip to each of the four other fingertips. The Lunar Lander game has a 2D continuous action space that controls the main engine and lateral thrusters (details in Appendix A.3). The interface $\pi_\theta$ is a linear transformation from the 4D hand pose command to the 2D thruster action space (i.e., $\theta$ consists of 8 parameters). The initial interface $\pi_{\theta_0}$ is a random linear function, and is usually difficult for the user to operate (e.g., because it requires making difficult hand gestures, or never fires the lateral thrusters). Lunar Lander is a challenging game, even when played with a joystick or keyboard. What makes it more challenging in this setting is that, unlike 2D cursor control, where there is an obvious prior mapping that moves the cursor in the direction of the mouse without any rotation, there is no obvious mapping from hand gestures to thruster actions that the user would have in mind.

The results in Fig. 4 show that, after 200 episodes of online training, MIMI learns an interface that enables the expert user to maneuver the lander, usually avoid crashing, and occasionally land successfully. The interface enables the user to fire the right thruster by touching their index finger to their thumb, fire the left thruster and main engine by touching their ring finger to their thumb, and fire the main engine by slightly curling all fingers. The 8 interface parameter values $\theta$ learned through MIMI would have been difficult to determine in advance, given uncertainty about the range of comfortable hand pose commands, and the non-obvious intuitiveness of using individual fingers for lateral thrusters vs. all fingers for the main engine. Interestingly, the final interface does not enable the user to fire the left thruster in isolation (i.e., all gestures that fire the left thruster also fire the main engine simultaneously). This turns out to be okay, since the user wants to descend, and alternating between firing the left thruster and main engine together then firing the right thruster without the

main engine is a reasonable strategy for descending without tilting. This would have been difficult to anticipate in advance, but nevertheless emerged through co-adaptation.

## 6 Discussion

We presented a proof of concept that, through unsupervised human-in-the-loop reinforcement learning, we can learn an assistive interface from scratch in less than 30 minutes without any explicit user feedback or prior knowledge of the user's desired tasks. Our experiments show that, for a variety of users (12 participants in our user study, and 12 in each of the offline evaluations), command modalities (high-dimensional eye gaze and hand gestures, and low-dimensional keyboard commands), environments (typing, simulated robotic control, playing a video game, and cursor control), and tasks (flipping light switches and reaching for bottles with the same simulated robotic arm), MIMI's mutual information objective can be used to rank interfaces without observing ground-truth rewards for the user's desired task.

### 6.1 Limitations

MIMI is limited by the fact that the correlation between the mutual information objective $\mathcal{I}(\mathbf{x}_t, (\mathbf{s}_t, \mathbf{s}_{t+\Delta}))$ and the ground-truth reward function can depend on the time offset $\Delta$ (Sec. 5.1). Since we do not assume access to ground-truth rewards, choosing the value of the hyperparameter $\Delta$ requires prior knowledge of the timescale of the user's desired control over the system (i.e., whether the user wants to be able to control what happens next, or what happens by the end of the episode).

The main limitation of our experiments is that the largest interface only has 8 parameters, due to constraints on the duration of a user study and the data efficiency of the Bayesian optimization algorithm we use for RL. One direction for future work is to instantiate MIMI with a data-efficient deep RL algorithm that scales to learning high-dimensional policies, such as REDQ [69] or policy evaluation networks [70].

### 6.2 Future Work

Another direction for future work is evaluating MIMI on real-world systems in which explicit user feedback is sparse or unavailable, such as brain-computer interfaces for users with total locked-in syndrome. MIMI could also be helpful for designing creative tools for artists and musicians where the 'task' or 'ground-truth reward function' is inherently difficult to specify, such as a tool that helps users navigate the high-dimensional latent space of a generative model of images using physical gestures and virtual reality [71], an instrument that provides users with a playable, low-dimensional latent space of music audio [72], or a tool that helps designers carve out discriminable gestures from low-dimensional embeddings [73].

There may be cases where the user can observe the state $\mathbf{s}_t$, but the interface cannot (e.g., the user can look around with their eyes, but the interface is not equipped with a camera). A useful extension of MIMI would extract implicit feedback from the stream of command signals $\mathbf{x}_t$ alone, without assuming access to the states $\mathbf{s}_t$.

Biological neurons maximize the mutual information between their inputs and outputs through local, unsupervised learning—known as the infomax principle [74]. Ideas from recent work on greedy, self-supervised representation learning based on the infomax principle [75] could be useful for scaling MIMI to deep neural network interfaces.

## 7 Acknowledgments

This work was supported by Berkeley Existential Risk Initiative, NSF IIS-1651843, ARL DCIST CRA W911NF-17-2-0181, Weill Neurohub, and NSF CAREER. Thanks to Natasha Jaques for giving feedback on this project during its early stages and pointing us to related work. Thanks to `lauren`, `Quantum1000`, `guillefix`, and the Transhumanists in VR community for many stimulating discussions about MIMI, human-computer interfaces, and multi-agent learning. Thanks to anonymous NeurIPS reviewers eRaa, phLR, and evLC for their constructive feedback.

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
