# OpenReview forum: "First Contact: Unsupervised Human-Machine Co-Adaptation via Mutual Information Maximization"
_NeurIPS.cc/2022/Conference — NeurIPS 2022 Accept_

### Official Review · Reviewer_evLC · 2022-06-22

**Rating:** 7
**Confidence:** 4
**Soundness:** 3 good
**Presentation:** 4 excellent
**Contribution:** 3 good

**Summary:**

This paper presents a reinforcement learning algorithm that uses mutual information as a proxy reward for achieving understanding between a computer with a randomly perturbed UI and a human trying to accomplish a task using that UI, without knowing how it is perturbed.  The only feedback (signal) to the algorithm is the entropy of the humans movements/inputs where it is assumed that the human would be making more random/chaotic inputs in a sense proportional to how unexpectedly (unintuitively) the game’s state changed with respect to the action taken by the user.  The main evaluation is with a space ship control game where the actual motion of the ship was perturbed/offset initially by some random theta degrees.  The algorithm eventually learns to reduce theta either to zero or -180 degrees so that it exactly follows the human’s command.  Only at these two stable points does the human stop acting so randomly in response to the state changes, since the state changes are, apparently and acceptable reaction to the action (so further exploration by the human is not required).

As the authors state: “the key idea  in this paper is that, regardless of the task, when an interface is more intuitive,
the user’s commands are less noisy.”  They formalize this “noise” as “the mutual information between
the user’s command signals and the induced state transitions in the environment.”  Their work is in line with recent literature in user empowerment where the machine does not assume or infer the users’ goals but only uses cues that indicate the user’s satisfaction with the result as a reward upon which to learn in this human-in-the-loop system.



**Questions:**

Out of curiosity did you consider any other metrics of noise assessment, e.g. standard deviation of responses, length of time of “attempted corrective” responses, etc. and how did these compare to mutual information?

Was there anything special about you users, were they graduate students in computer science, friends, co-workers? Were they motivated in some way to complete the task?   I can imagine the “average” user simply abandoning a malfunctioning interface.


**Limitations:**

This contribution is limited to the space of adaptive interfaces where the correct settings for the user’s task cannot be known in advance but where the parameters of the controlling algorithm have the ability to adapt themselves to a more optimal setting.  The assumptions are reasonable but not exhaustively studied.  The process of adaptation is comparatively slow compared to other methods, so a stronger motivating use case would have been better.  The method could have been more exhaustively compared versus other human-in the loop methods such as NLP or facial expression analysis or simply giving the user an explicit “performance feedback” channel on the algorithm’s performance (e.g. 1 to 10 as a keyboard input) and evaluated with respect to speed of convergence and user satisfaction (e.g. if could take longer yet users might prefer it as more natural.



**Strengths And Weaknesses:**

The key strength of the paper is that it is interesting and the idea that users will express frustration with a poorly performing interface by random frantic exploration of potential alternative commands that might work seems plausible and is supported by the authors first study that shows that their mutual information pseudo-reward is correlated with interfaces that adapt to meet the users expectations (become more intuitive)  As the authors state: “the key idea  in this paper is that, regardless of the task, when an interface is more intuitive,
the user’s commands are less noisy.”  The authors give reasonable proof of the fact that with adaptive interfaces, users do seem to exhibit the behavior of taking more noisy actions when the resulting effects in the environment (state changes) are not what would be intuitively expected.  The authors do include a user study with a simple adaptive interface that shows that their mutual information pseudo-reward can drive convergence in this case.  The authors also show that a knowledgeable user can drive adaptation using the pseudo-reward.

As the author states there has been a number of attempts to find “natural” reward signals in the real world when interacting with humans whether these be through NLP, BCI (EEG), facial expressions and physiological signals.  None of these are as easy to obtain as the “noise” in the input that these authors leverage, making this perhaps the most useable of all prior proposed methods in the field.

The weakness of the paper is that the use case is not immediately compelling.  The first contact scenario is a good framing for this, however unlikely, but in reality, this is a more theoretical method for machines to learn preferences from humans.  The more compelling use case is probably in applying this as a feedback mechanism to search engines or to personalize the “fit” of something like exoskeletons or prosthetic limbs.  The user studies are not exhaustive, but I would not expect that for a contribution that is fundamentally about the algorithm.  It is laudable that the authors disclosed that the singular subject in the feasibility study was the first author and  while on the one hand it could be argued that having such an informed participant as an evaluator made the task “easier” for the machine (potentially the author knew to proportionally vary the randomness of the responses in a way that would facilitate convergence) it is not entirely to be dismissed especially in considering a case where a person was trying to “fit” or “train” a machine intentionally knowing how the machine learned through randomness.  The authors show that this is a naturalistic response in general, so intensifying it intentionally would not be entirely out of line with user behaviour.

Also, I believe the term “co-adaptive” in the title is a bit misleading as you do mention that the algorithm has no ability to adapt the behavior of the human.  I believe you are saying that the human naturally seems to adapt their behavior in response to the algorithms “correctness” and that you are relying on the existence of this adaptation otherwise you pseudo reward would not work, but maybe clarifying this a bit would be good.

---

> ### Author Response · Authors · 2022-07-27
> **Response to reviewer evLC**
>
> Thank you for the thoughtful and constructive review.
>
> > Out of curiosity did you consider any other metrics of noise assessment, e.g. standard deviation of responses, length of time of “attempted corrective” responses, etc. and how did these compare to mutual information?
>
> In this paper, we did not consider any other metrics of noise assessment, as they would have been less general than our mutual information score. For example, there may be some domains in which the task requires that the user provide command signals with a large standard deviation, even when the interface is intuitive and maximizes mutual information. Furthermore, in order to detect corrective command signals, we would need to make assumptions about what kinds of user command signals correspond to corrections vs. progressing on the task. Our goal in this paper is to propose a general method for interface optimization. That being said, we think there is room for future work on more domain-specific objectives.
>
> > Was there anything special about you users, were they graduate students in computer science, friends, co-workers? Were they motivated in some way to complete the task? I can imagine the “average” user simply abandoning a malfunctioning interface.
>
> Thank you for raising this good point. The participants were primarily graduate students in computer science and friends. As mentioned in Appendix A.1, they were each compensated with a $10 Amazon gift card. To your larger question, we intend MIMI for users who are motivated to practice and co-adapt with the interface, even if the interface is initially unintuitive and unresponsive. For example, consider a locked-in, paralyzed patient who may be willing to spend hours, days, or even weeks training their communication interface by interacting with it.

---

### Official Review · Reviewer_phLr · 2022-07-12

**Rating:** 5
**Confidence:** 3
**Ethics Flag:** Yes
**Soundness:** 3 good
**Presentation:** 2 fair
**Contribution:** 3 good

**Summary:**

This paper presents a method for building an adaptive user interface in an unsupervised manner. The key idea is to use the mutual information between the user input and the state transition. Based on this idea, the authors propose an algorithm to maximize the mutual information lower bound from a small amount of user operation data. The proposed approach is evaluated using five existing dataset, real-world user study data, and another expert user demonstration.

**Questions:**

- I would like a more detailed explanation and discussion on the last point I raised above. Are there any results that suggest that the proposed method can complete the desired task more quickly or with greater user satisfaction than a scenario in which users continue to use a non-adaptive interface and become proficient with it?

**Ethics Review Area:**

["Responsible Research Practice (e.g., IRB, documentation, research ethics)"]

**Limitations:**

- There is an experimental discussion on interfaces where the proposed method cannot be successfully applied, which is a good point for the paper.
- Although the study involved numerous user experiments, there was no specific mention of a process for obtaining consent from participants or for ethical review.

**Strengths And Weaknesses:**

### Strength
- The unsupervised user adaptation task setting is interesting, and the idea of using mutual information content seems reasonable.
- Experimental evaluations have been conducted from multiple perspectives, demonstrating at a minimum the effectiveness of the proposed method.

### Weaknesses
- While offline evaluation using existing data makes sense, evaluating the actual use of adaptively changing interfaces with this approach is not necessarily sufficient. The cursor control task seems a bit easier than the "unknown" interface that this study assumes. Multiple user experiments on diverse interfaces with similar or greater participants would increase the method's reliability. The author's demonstration is somewhat unreliable as experimental evidence.
- As for the method's claim that it does not require prior knowledge about the desired task, it does not appear to be strictly substantiated, as there are no corresponding experiments.
- Considering the purpose of the study, what should be shown is the advantage of the system compared to the user's solo adaptation to the system. However, the experiment mainly only compares the initial state and after adaptation, and there is no discussion of what happens after the participants use the baseline interface for an extended period.

---

> ### Author Response · Authors · 2022-07-27
> **Response to reviewer phLr**
>
> Thank you for the thoughtful and constructive review.
>
> > Are there any results that suggest that the proposed method can complete the desired task more quickly or with greater user satisfaction than a scenario in which users continue to use a non-adaptive interface and become proficient with it?
>
> We have added Figure 7 to the appendix, in which we present a view of the data from the user study in Figure 3 that illustrates how a user adapts to a fixed interface. During the user study, the user is repeatedly presented with a new interface and asked to operate it for 10 episodes. During those 10 episodes, the user becomes more proficient at using that fixed interface. When presented with a random interface, the user takes a longer time to learn to use the interface and achieves worse final performance, compared to when they are presented with an interface that is being optimized by MIMI.
>
> > Although the study involved numerous user experiments, there was no specific mention of a process for obtaining consent from participants or for ethical review.
>
> We obtained informed consent from each participant, as well as IRB approval for our study. We have added this to Appendix A.1.

---

### Official Review · Reviewer_eRaa · 2022-07-26

**Rating:** 5
**Confidence:** 3
**Soundness:** 2 fair
**Presentation:** 3 good
**Contribution:** 3 good

**Summary:**

This paper aims to solve the problem of human-machine interface with supervision or prior knowledge about the user’s desired tasks. This is challenging due to the large space of interface designs. This paper proposes a reinforcement-based interface update method by maximizing the mutual information between the user’s commands and the induced state transition, where an underlying key idea is that “the more intuitive the interface, the less noisy the user’s commands.” The main contribution is the design of mutual information objective/reward for enhancing interfaces. The experiment section also shows several studies to demonstrate the effectiveness of the updated interfaces and application examples.

**Questions:**

- It would be great why algorithm 1 uses a validation set for the return computation. Further, in this work, the authors can obtain a small amount of data. In general, the size of the validation set will be smaller than that of the training set. How does it affect the performance?

- Figure 2 requires clarifications. First, it is hard to see the correlation between mutual information and true reward in Fig. 2(a)-(c). Although Fig. 2(e) and (f) show the Spearman correlation coefficient (i.e., 0.43 and 0.69), it is not clear how much the value is significant to say the two values of information are correlated.

- Second, Fig. 2(d) shows a large number of time offsets that may result in very long time gaps between observed states. It should clarify how it affects the data collection, estimator update, etc considering the real-time human-machine interaction.

- Any comparison with baselines will be helpful in understanding and also support the proposed method.


**Strengths And Weaknesses:**

The main strength of this paper is that the intuitive design of mutual information rewards considering the user’s influence on the next state and its opposite.

The major weakness of this paper is there is no comparison with baselines. It is unclear whether the given dataset/problem is easy to design interface. Further, it is still hard to clearly confirm the correlation between the mutual information and true reward over results, particularly in Fig. 2.

---

> ### Author Response · Authors · 2022-07-27
> **Response to reviewer eRaa: new Lunar Lander experiments, new analysis of user adaptation to random interfaces baseline**
>
> Thank you for the thoughtful and constructive review. To address concerns regarding the strength of the random interfaces baseline, we have added an analysis of user adaptation to the random interfaces baseline to the appendix, and added new trials to the Lunar Lander experiment in the main paper. In the following paragraphs, we address each of your questions in more detail. Please let us know whether our responses sufficiently address your concerns, or if we can clarify anything further.
>
> > Any comparison with baselines will be helpful in understanding and also support the proposed method.
>
> To our knowledge, there are no prior methods that are capable of learning an interface in a completely unsupervised manner. Hence, we compare our method to the nearest alternative method that shares the same assumptions as our method (no prior knowledge of the task or direct supervision): a random interface. The protocol for sampling and evaluating random interfaces is outlined in paragraph 2 of Section 5.2.
>
> To further clarify the difference between MIMI and the random interfaces baseline, we have added Figure 7 to the appendix, in which we visualize to what extent the user adapts to MIMI vs. random interfaces. During the user study, the user is repeatedly presented with a new interface and asked to operate it for 10 episodes. During those 10 episodes, the user becomes more proficient at using that fixed interface. When presented with a random interface, the user takes a longer time to learn to use the interface and achieves worse final performance, compared to when they are presented with an interface that is being optimized by MIMI.
>
> We have also added new trials with two more random seeds to the Lunar Lander experiment in Figure 4, which show that MIMI enables the user to achieve substantially higher landing success rate than the random interfaces baseline.
>
> > it is not clear how much the value is significant to say the two values of information are correlated.
>
> Thank you for raising this good point. As the rank correlation between the mutual information score and ground-truth task reward increases, then selecting the interface with the highest mutual information score will enable the user to obtain higher ground-truth task reward as well. Indeed, Figure 5 in the appendix shows that, in all 5 datasets from Figure 2, the interface with the highest mutual information score is also the interface that enables users to achieve the highest ground-truth reward. We will emphasize this more in paragraph 2 of Section 5.1.
>
> > It would be great why algorithm 1 uses a validation set for the return computation.
>
> As per the [Data-Efficient Mutual Information Neural Estimator](https://arxiv.org/abs/1905.03319) paper cited in Section 2.2, using the validation set improves the data efficiency of the mutual information lower bound estimator. Although the validation set is smaller than the training set, we find that the resulting mutual information scores are effective for online optimization of an interface (Sections 5.2 and 5.3), and that the mutual information scores correlate positively with ground-truth task rewards from offline data (Section 5.1).
>
> > Second, Fig. 2(d) shows a large number of time offsets that may result in very long time gaps between observed states. It should clarify how it affects the data collection, estimator update, etc considering the real-time human-machine interaction.
>
> We apologize for the miscommunication. The time offset \Delta does not affect the data collection, estimator update frequency, or real-time human-machine interaction. Rather, it is a hyperparameter that controls how we compute the mutual information score after each round of data collection with a new interface (lines 8-9 in Algorithm 1). We elaborate in paragraphs 3 and 4 of Section 5.1.
>
> > First, it is hard to see the correlation between mutual information and true reward in Fig. 2(a)-(c)
>
> In the title of each plot in Figures 2(a)-(c), we display the Spearman rank correlation coefficient \rho, which ranges from -1 (perfect negative correlation) to 0 (no correlation) to +1 (perfect positive correlation). The correlation coefficients for Figures 2(a)-(c) are 0.45, 0.50, and 0.11 respectively.

---

> > ### Comment · Reviewer_eRaa · 2022-08-02
> > **Unsupervised Co-Adaptation**
> >
> > >To our knowledge, there are no prior methods that are capable of learning an interface in a completely unsupervised manner. Hence, we compare our method to the nearest alternative method that shares the same assumptions as our method (no prior knowledge of the task or direct supervision): a random interface. The protocol for sampling and evaluating random interfaces is outlined in paragraph 2 of Section 5.2.
> >
> > I am not familiar with human-machine interaction, but I could easily search "unsupervised co-adaptation" related papers. For example, I could easily find an unsupervised co-adaptation paper [1] using the keywords though the paper may not be a suitable baseline. However, my point is that unsupervised multi-agent learning/co-training/co-adaptation/human-machine(robot) interaction is not a new area. The comparison with a random interface does not support if the proposed approach is really good enough. I think any ablation studies, comparisons with heuristic methods, user studies, etc are necessary to support the validity of the proposed approach as well as evaluation.
> >
> > [1] D. De Santis, P. Dzialecka and F. A. Mussa-Ivaldi, "Unsupervised Coadaptation of an Assistive Interface to Facilitate Sensorimotor Learning of Redundant Control," 2018 7th IEEE International Conference on Biomedical Robotics and Biomechatronics (Biorob), 2018.
> >
> >
> > > In the title of each plot in Figures 2(a)-(c), we display the Spearman rank correlation coefficient \rho, which ranges from -1 (perfect negative correlation) to 0 (no correlation) to +1 (perfect positive correlation). The correlation coefficients for Figures 2(a)-(c) are 0.45, 0.50, and 0.11 respectively.
> >
> > My question was it's hard to interpret Fig. 2(a)-(c). In particular, from (a) and (b), I cannot see any pattern of points. Further, it seems there is no difference between methods.

---

> > > ### Author Response · Authors · 2022-08-05
> > > **Additional comparison and analysis**
> > >
> > > > I am not familiar with human-machine interaction, but I could easily search "unsupervised co-adaptation" related papers. For example, I could easily find an unsupervised co-adaptation paper [1] using the keywords though the paper may not be a suitable baseline. However, my point is that unsupervised multi-agent learning/co-training/co-adaptation/human-machine(robot) interaction is not a new area. The comparison with a random interface does not support if the proposed approach is really good enough. I think any ablation studies, comparisons with heuristic methods, user studies, etc are necessary to support the validity of the proposed approach as well as evaluation.
> > >
> > > Thank you for raising the De Santis et al. 2018 paper to our attention. We have added it to the related work section. The method proposed in that paper applies principal component analysis to a dataset of command signals to fit a linear interface that maps commands to actions. We have applied this method to the data from our cursor control user study, and found that it learns an unintuitive interface: the learned interface rotates the user's mouse vector by 65 degrees, whereas MIMI learns to rotate either 0 degrees or 180 degrees. In our user study, users who operated an interface that rotated their mouse by an angle between 0 and 180 performed substantially worse than users who operated an interface that rotated their mouse either 0 or 180 degrees. Hence, MIMI learns an interface that performs better than the interface learned by the prior method.
> > >
> > > The key difference between MIMI and the prior method is that the prior method does not necessarily maximize the intuitiveness of the interface (it simply applies PCA to the command signals), whereas MIMI maximizes the intuitiveness of the interface via the $\mathcal{I}(\mathbf{s}_t, \mathbf{x}_t)$ mutual information term.
> > >
> > > > My question was it's hard to interpret Fig. 2(a)-(c). In particular, from (a) and (b), I cannot see any pattern of points.
> > >
> > > We apologize for the lack of clarity. We have added regression lines to Fig. 2 to visually illustrate the rank correlation between our mutual information scores and the ground-truth task rewards.
> > >
> > > > Further, it seems there is no difference between methods.
> > >
> > > Fig. 5 in the appendix illustrates the differences between methods. We will move this figure into the main paper for the final version.
> > >
> > > Please let us know if there are any other comparisons or analyses that would be helpful!

---

> ### Author Response · Authors · 2022-07-31
> **Request for feedback**
>
> Thank you again for the detailed review. We would like to know if our rebuttal adequately addressed your concerns. Is there anything else we can clarify or address?

---

### Review · Ethics_Reviewer_eHJT · 2022-08-02

**Recommendation:**

General recommendations are not needed in this paper. To anticipate future potential ethical concerns and avoid an ethics flag, always add all documents related to research ethics practices in the appendix.

**Ethics Review:**

The paper raises no particular ethical issues.

---

### Review · Ethics_Reviewer_ikni · 2022-08-08

**Recommendation:**

In Appendix A1 the authors can consider adding the average duration of the user study so the compensation could be better evaluated.

**Ethics Review:**

This paper presents a novel interface that can translate users’ raw command signals into robot actions without the requirement of prior mapping. The authors evaluated the method through both evaluation of prior datasets and observational study with 12 participants. The authors have added details of the user study and IRB information in the appendix. There are no additional ethics issues.

---

### Meta-Review · Area_Chair_5hw6 · 2022-08-25

**Recommendation:** Accept
**Confidence:** Less certain

**Metareview:**

The paper describes an approach to learning an adaptive user interface (i.e., mapping raw inputs to the agent's actions) in an unsupervised way via reinforcement learning. The goal is to learn interfaces that are intuitive for the user, with the supposition that the user's inputs become less noisy as the interface becomes more intuitive. To that end, the proposal is to use the mutual information between the raw input provided by the user and the resulting state transitions as a reward proxy. The approach is evaluated on a series of control and typing domains as well as a small-scale user study involving a cursor control task.

The paper was reviewed by three researchers who read the author response an discussed the paper with the AC. The reviewers agree that the problem of adapting a user interface in an unsupervised way is interesting and the proposed use of mutual information for adaptation is sensible and interesting. The reviewers initially raised concerns about the absence of a compelling use case and the experimental evaluations, notably the lack of appropriate baselines (Reviewer eRaa) and inadequate experiments (Reviewers eRaa and phLr). The authors made a concerted effort to address most of the reviewers' concerns, which included experiments conduced on the cursor domain using the alternative method suggested by Reviewer eRaa. However, the authors did not address the experimentation issues raised by Reviewer phLr, who finds that the paper lacks experimental evidence for some of the claims being made. As it stands, the paper doesn't show that the interface that can be achieved with this approach is truly intuitive. Making such a claim requires comparative experiments with appropriate baseline interfaces and more detailed user analyses. As such a detailed set of user studies may out-of-scope for a conference-length algorithms paper focused on the use of mutual information as a reward proxy for interface learning, the claims in the paper should be revisited.

**Award:**

No

---

### Decision · Program_Chairs · 2022-09-14

Accept